# Spatio-Temporal Patterns of the SARS-CoV-2 Epidemic in Germany

**DOI:** 10.3390/e25081137

**Published:** 2023-07-29

**Authors:** Hans H. Diebner

**Affiliations:** Department of Medical Informatics, Biometry and Epidemiology, Ruhr-Universität Bochum, 44780 Bochum, Germany; hans.diebner@rub.de

**Keywords:** COVID-19, SARS-CoV-2, epidemic spatial heterogeneity, diversity, spatio-temporal patterns, cluster analysis

## Abstract

Results from an explorative study revealing spatio-temporal patterns of the SARS-CoV-2/ COVID-19 epidemic in Germany are presented. We dispense with contestable model assumptions and show the intrinsic spatio-temporal patterns of the epidemic dynamics. The analysis is based on COVID-19 incidence data, which are age-stratified and spatially resolved at the county level, provided by the Federal Government’s Public Health Institute of Germany (RKI) for public use. Although the 400 county-related incidence time series shows enormous heterogeneity, both with respect to temporal features as well as spatial distributions, the counties’ incidence curves organise into well-distinguished clusters that coincide with East and West Germany. The analysis is based on dimensionality reduction, multidimensional scaling, network analysis, and diversity measures. Dynamical changes are captured by means of difference-in-difference methods, which are related to fold changes of the effective reproduction numbers. The age-related dynamical patterns suggest a considerably stronger impact of children, adolescents and seniors on the epidemic activity than previously expected. Besides these concrete interpretations, the work mainly aims at providing an atlas for spatio-temporal patterns of the epidemic, which serves as a basis to be further explored with the expertise of different disciplines, particularly sociology and policy makers. The study should also be understood as a methodological contribution to getting a handle on the unusual complexity of the COVID-19 pandemic.

## 1. Introduction

For about 3 years, beginning in September 2019, the SARS-CoV-2/COVID-19 pandemic shook the world in an almost unprecedented way [1]. Three years after the outbreak, it is still not clear whether the pandemic entered an endemic phase and how long a crisis-like situation will persist or re-emerge [2,3]. There are still too many unknowns to be able to provide clear prognoses, although the flood of COVID-19-related publications is without example [4,5]. Equally unprecedented is the fact that a high proportion of the literature dealing with the pandemic is meta-scientific and/or meta-bibliographic in nature or belongs to the sociology of behaviour or the sociology of science. Topics include discussions of malicious or accidental miscommunication even within the scientific context [6]. Contributions address misconduct but are themselves often characterized by sheer polemic, if not denunciation. This fact may best be summarised that we face a harsh “COVID-19 infodemic” [7]. It is no surprise that a number of surveys and studies clearly point to a polarisation and radicalisation in public attitudes and behaviour driven by a polarisation in elite rhetoric that hinders effective responses to the COVID-19 crisis [8] and gives rise to an increasing social Darwinism [9]. Thus, health behaviour is increasingly driven by political ideology [10,11,12], such that the resulting epidemic dynamics have become almost unpredictable and uncontrollable.

Several attempts have been made to describe the complex spread dynamics of SARS-CoV-2 infections, both purely temporally and spatially [13], using deterministic compartmental models, as well as stochastic and neural network-based models [14]. Some of the models have a considerable degree of complexity and, at least at the beginning of the pandemic, were attributed “life and death importance” [15]. For example, attempts were also made to include the dynamics of containment measures (cf. [16]), vaccination coverage [17,18,19], as well as the dynamics of information propagation [20] in the model. As an alternative to these mathematical modelling approaches, various statistical modelling approaches were also applied. Regressions of both spatio-temporal incidence and fatality rates on socioeconomic and other independent (predictor) variables, such as vaccination coverage or containment measures have been addressed. Using a least absolute shrinkage and selection operator (LASSO) regression, a U.S. study [21] identified subsets of socioeconomic predictors that are indicators of vulnerable counties. Another study conducted in Germany [12] also examined the impact of socioeconomic determinants at the local level on the basis of linear mixed models, but in addition, also examined the impact of political predictors, i.e., party vote shares. However, all the mathematical modelling approaches and statistical regressions mentioned, as well as the majority of similar modelling articles impossible to list here, depend on controversial assumptions.

Although no opposition to the various modelling approaches is intended here, the mixed situation calls for complementary solutions. The analysis presented is restricted to the epidemic in Germany. We continue to assume that the complex set of causes and associations should be reflected in the spatio-temporal patterns of infection dynamics. However, we refrain from regression and modelling assumptions, i.e., we look for dynamic patterns at both temporal and spatial levels in the data themselves. In this way, it is intended to provide an atlas that offers clues to possible influencing factors. Although some interpretations that are important to us (or obvious) are suggested about the determinants of the observed patterns, overall we are cautious about interpretations. In this way, we refrain from value-based assumptions and preconceived hypotheses and instead offer a basis for generating valid hypotheses with this explorative approach.

## 2. Materials and Methods

### 2.1. Methodological Basis

In this paper, we take a closer look at the spatio-temporal dynamics of the epidemic in Germany. Due to the aforementioned socio-behavioural imponderabilities, the problem is inherently systemic and adaptive in that sense, that preventive measures, including their associated compliances and epidemic activity, are bidirectionally related via nonlinear feedback loops [22]. It follows, even if containment measures were precisely datable, one could not rule out the possibility that they would ultimately be counteracted and even be changed to the opposite. The notation of a “self-disorganisation” suggests itself. We here take the stance that the recorded, more or less objective incidences, should speak for themselves since most of the aforementioned socio-behavioural determinants defy quantifiability. We aim at presenting suitably quantified spatio-temporal patterns of the German epidemic activity in terms of features of a weekly recorded incidence time series at the relatively fine-grained spatial resolution of German districts. In essence, our approach is explorative in that we invert the search direction: striking spatio-temporal incidence patterns provide timestamps and spatial clues that point to external determinants that trigger changes and allow for associations with prevailing social conditions.

Specifically, we derive diversity measures based on entropy calculations, but also apply more recently introduced methods from the field of network analysis. These include methods of dimensionality reduction, multidimensional scaling, as well as cluster analyses. Taken together, we try to provide as comprehensive a picture of the complexity of the pandemic as possible, using only available incidence data. Some facets of the complexity of the COVID-19 epidemic have been published recently [23,24] and we ask the readers to combine these previous results mainly focusing on temporal features with the new findings presented here, which put more emphasis on spatial patterns.

Throughout the paper, statistical calculations and the creation of graphs were completed using R version 4.3.0 [25]. The used R program packages are mentioned at the appropriate places.

### 2.2. General Settings and Nomenclature

In the following, contextually, either the 400 German rural/urban districts (counties) or the 16 federal states are labelled by index i=1,…,400 or i=1,…,16, respectively. In graphs, however, federal states are represented by the official two-letter abbreviation (see the list of abbreviations in the appendix). The age-specific incidences,
(1)$$I_{i}\left(a,t\right)=\frac{counts_{i}\left(a,t\right)}{popsize_{i}\left(a\right)},$$
given by registered counts, $counts_{i}\left(a,t\right)$ per sub-population size, $popsize_{i}\left(a\right)$, at time point *t* of counties i=1,…,400 (or, depending on context, federal states i=1,…,16) have been calculated from the counts retrieved from the Robert Koch-Institute (RKI) database [26] and from the respective age-specific sub-population sizes retrieved from [27]. Depending on the context, age *a* either refers to the age classes (in years) [0,5), [5,12), [12,18), [18,30), [30,40),…,[70,80), $80^{+}$ or to kids:= [0,18), adults: =[18,60), $seniors:\;=60^{+}$, and time *t* is given by the calendar date in weekly steps from January 2020 through the end of August 2022, i.e., we have t=1,…,135 time steps. Of note, Berlin is sub-divided into 12 administrative districts. Although COVID-19 counts are separately listed in the RKI database with respect to these districts, we use aggregated data for Berlin constrained by the database structure of the Census Bureau. Berlin, as well as Hamburg, thus function both as a single county and as a federal state.

### 2.3. t-sne

The “t-distributed stochastic neighbour embedding”, in short, t-sne, is a commonly applied dimensionality reduction algorithm [28] with a precursor, called “SNE”, introduced by [29]. For a brief outline, assume that each of the 400 incidence time series corresponding to the 400 German districts is represented as a point in a 135-dimensional vector space with 135 being the length of the weekly sampled time series. SNE, and likewise, t-sne, as applied here, reduces the 135-dimensional to a two-dimensional vector space while preserving the proximity of data points based on a distance or similarity measure. Such a reduction in dimensionality needs to specify a so-called perplexity parameter, which, in a nutshell, reflects the users’ taste of how quickly similarity should fade out with increasing distance, i.e., which area is conceived as neighbourhood. This trick allows for a visual inspection of clustering patterns in two dimensions, if any. Calculations are based on algorithms provided by [30].

### 2.4. UMAP and PCA

“Uniform manifold approximation and projection”, in short, UMAP, has been introduced by [31] as a dimensionality reduction algorithm that out-performs t-sne by means of preserving global structures, at least such is the claim. However, comparative studies do not allow a definitive conclusion. Analogous to t-sne, UMAP necessitates a free parameter, called *n*-neighbors, to be set, which defines a custom range of neighbourhoods. Both algorithms compete with the well-known method of principal component decomposition/analysis (PCA) [32], which seeks an optimal explanation of variability after the decomposition while preserving the overall variance in the data. We present the results from applications to the COVID-19 time series of all three algorithms side by side and conceive this as a sensitivity analysis. The used algorithms are provided by [33,34]. The PCA algorithm is endowed with the possibility to calculate normal data ellipses around a set of predefined data points, which are assumed to constitute clusters. Therefore, an approximate (pseudo) confidence measure for the separability of clusters is available.

### 2.5. Correlation Matrix and Hierarchical Clustering

Calculating pairwise correlation coefficients of the 16 federal state-specific incidence time series can be conceived as a reduction to a one-dimensional space since correlation is just a specific similarity measure. Along with hierarchical clustering, a visualisation of the correlation matrix should essentially yield the same information as a two-dimensional reduction, as long as specific structures that can only be recognised by a specific algorithm are absent. Both Pearson and Kendall correlations are calculated since it is commonplace. Pearson is good at recognising linear correlations whereas Kendall can also be applied to nonlinearly correlated data vectors. Thus, a comparison of the results after the application of all suggested methods will decidedly bring added value in interpreting the dynamic hallmarks of the epidemic. The correlation plots are created using the R package provided by [35]. Hierarchical cluster analysis is based on Ward’s minimum variance method, which aims at finding compact, spherical clusters. Specifically, we use the “ward.D2” method, which means that dissimilarities are squared before cluster updating, according to the package manual.

### 2.6. Multidimensional Scaling and Network Graphs

Multidimensional scaling (MDS) is the umbrella term of a family of dimensionality reduction algorithms. MDS aims at preserving distances, or, conversely, proximities between data points. Note that this differs from the t-sne neighbourhood embedding, which clusters neighboured points tightly in order to clearly and visually separate the clusters. Here, we exclusively use non-metric MDS based on spline transformations. In other words, the spline function *f* transforms dissimilarities $\delta_{ij}$ to disparities ${\hat{d}}_{ij}$ via ${\hat{d}}_{ij}=f\left(\delta_{ij}\right)$. Corresponding dissimilarities in the low-dimensional space are then found by minimising a so-called stress function, which, in essence, is a function of the difference between disparities in the original and reduced spaces. For details, cf. [36].

Recently, MDS has been combined with graph visualisation with increasing popularity. Based on a scalar measure of proximity between any pair of data points (here time series), such as, e.g., correlation coefficients, the data points can still be represented in two dimensions in the form of nodes (or vertices), where the scalar similarities determine the strength of the connecting edges. The spatial arrangement of the nodes can then be based on similarities calculated by minimising the corresponding stress function. Other, usually ambiguous arrangements are in use, which can be constrained by the requirement of having non-overlapping nodes. Also popular is the simple arrangement of nodes into a circle, whereby nodes assumed to share a cluster can be arranged such that they are adjacently located on the circumference.

Although this type of presentation in the form of a graph visualisation is suggestive to the eye of the beholder and therefore prone to misinterpretation, it can flank the exploratory approach if one is aware of the pitfalls. We present graphs which are based on correlation coefficients or on distances supplied by a PCA. Moreover, after the application of a Gaussian graphical model using LASSO, a partial correlation network can be derived, which reduces the number and strengths of the edges of the graph to a relevant magnitude. Please cf. [36,37], for a detailed description of the graph visualisation methods used here, including the Gaussian graphical model, referred to as “graphical LASSO”.

Network graphs are commonly published together with so-called centrality indicators. Network strength, sometimes also referred to as degree centrality, assigns an importance score to each node/vertex, which is, in our application, the sum of pairwise absolute values of correlations to all other nodes. Betweenness centrality measures quantify how strong given nodes build bridges between other pairs of nodes. Closeness centrality scores each node based on its strength to all other nodes in the network. It is worth noting that betweenness and closeness do not differ strongly from degree centrality if the similarity of the nodes is measured in terms of correlations. However, we will report these centrality measures for the sake of completeness. Expected influence, occasionally called eigen centrality, measures a node’s influence on the entire network. If correlations are used for the quantification of similarities, the expected influence differs from network strength only if positive and negative correlations are simultaneously present in the network since expected influence does not use absolute values when summing up the correlations.

### 2.7. Spatial Heterogeneity

Spatial Shannon entropy at time *t* is given by
(2)$$S\left(a,t\right)=\sum_{i=1}^{400}\frac{I_{i}\left(a,t\right)}{\sum_{j=1}^{400}I_{j}\left(a,t\right)}\ln\left(\frac{I_{i}\left(a,t\right)}{\sum_{j=1}^{400}I_{j}\left(a,t\right)}\right),$$
from which a measure of diversity (or spatial homogeneity), given by
(3)$$D\left(a,t\right)=\frac{e^{-S(a,t)}}{400},\;\mathrm{with}\;0\leq D\left(a,t\right)\leq 1,$$
can be calculated. The upper limit of 400 in the summation, as well as the denominator in the definition of diversity, Equation (3), refers to the number of districts and ensures diversity is a standardised measure. However, in order to detect possible differences between East and West Germany, the summation is also restricted to either the 75 East German or 325 West German districts. The proper denominators in Equation (3) scale the according diversity functions for East and West Germany and render them comparable.

For details on interpreting the diversity function, confer [38], and for an analogous application within an epidemiological context, see [39]. Briefly, for a given age class *a* at time point *t*, equal incidences over all counties give maximum entropy, hence maximum diversity D(a,t)=1. However, generally, D(a,t)<1 since synchronisation of the epidemic activity across districts appears to be unlikely. Particularly at the beginning of the epidemic with one or a few number of early index cases located within one or a few number of districts, D(a,t) will be close to 0. Over the course of time, intervals with a more or less homogeneous distribution of incidences across counties will probably alternate with asynchronous epidemic activities, as a consequence of spatially unequally distributed index cases of new epidemic waves and differences in socio-behavioural conditions. Here, we focus on trends and abstain from presenting statistical significance, i.e., confidence intervals for D(a,t) are ignored due to still controversial debates about their theoretical foundation [40]. In this regard, the reported incidence data may substantially deviate from true incidences such that confidence intervals would give rise to spurious certainty. Obviously, 1−D serves as a measure of heterogeneity. Thus, to avoid cumbersome formulations, it is sufficient to refer to the variation of *D* to indicate either homogeneity or heterogeneity depending on the context since it is the relations of *D* with respect to different regions that matter.

The following measure serves to capture changes in the dynamics of the epidemic in district/federal state *i*,
(4)$$\Delta \Delta I_{i}\left(t\right)=\frac{\frac{I_{i}\left(t\right)}{I_{i}\left(t-1\right)}}{\frac{I_{i}\left(t-1\right)}{I_{i}\left(t-2\right)}},$$
i.e., fold changes of weekly fold changes in incidence, which has also been used in [41]. Equation (4) goes to show the similarity to the weekly fold change in the effective reproduction number $R_{i}\left(t\right)/R_{i}\left(t-1\right)$ of area *i*. Taking the logarithm yields
(5)$$\ln\left(\Delta \Delta I_{i}\left(t\right)\right)=\ln\left(I_{i}\left(t\right)\right)-2\ln\left(I_{i}\left(t-1\right)\right)+\ln\left(I_{i}\left(t-2\right)\right),$$
which is used in the following due to its favourable symmetry with respect to zero. For convenience, $\ln\left(R_{i}\left(t\right)/R_{i}\left(t-1\right)\right)$ and $\ln\left(\Delta \Delta I_{i}\left(t\right)\right)$ are used interchangeably, whereby the first version can straightforwardly be extended to comparisons of two areas *i* and *j* at a given point in time, i.e., $\ln\left(R_{i}\left(t\right)/R_{j}\left(t\right)\right)$ (for details, cf. Equation (7) below).

With the exception of taking logarithms, Equation (5) bears resemblance to the so-called difference-in-difference method, which has frequently been used to identify causal effects of COVID-19 non-pharmaceutical interventions (cf. [42] for a review of the method and [43] for a systematic review of applications within the scope of COVID-19). Within the latter context, the counterfactual difference-in-difference method is applied to a setting which is assumed to be quasi-experimental in nature. If ${\bar{I}}_{i}^{pre}$ and ${\bar{I}}_{i}^{post}$ denote average incidences taken over a period before or after a containment measure has been mandated in district/federal state *i*, respectively, then
(6)$$\hat{\beta }=\left({\bar{I}}_{i}^{pre}-{\bar{I}}_{i}^{post}\right)-\left({\bar{I}}_{j}^{pre}-{\bar{I}}_{j}^{post}\right)$$
measures the effect of the containment action when *j* refers to a district/federal state without a corresponding mandate. Of note, Equation (6) does yield a reliable result if and only if areas *i* and *j* are “structurally” comparable, i.e., if a common trend assumption (constant underlying differences) holds (for details, cf. [42]).

Supposedly, the individual counties exhibit individual epidemic dynamics, in particular, as a consequence of different (starting and stopping of) containment strategies, but also due to inherent socio-structural conditions, hence creating and amplifying spatial heterogeneity. Equation (5), therefore, serves as an auto-difference-in-difference method to detect dynamical change points. It appears plausible that a district remains structurally “self-similar” over time, such that the auto-difference-in-difference is even more valid than the between-counties counterpart.

Specifically, as a consequence of the previous remarks, a special application of Equation (5) reads
(7)$$\ln\left(\frac{R_{i}\left(t\right)}{R_{j}\left(t\right)}\right)=\left[\ln\left(I_{i}\left(t\right)\right)-\ln\left(I_{i}\left(t-1\right)\right)\right]-\left[\ln\left(I_{j}\left(t\right)\right)-\ln\left(I_{j}\left(t-1\right)\right)\right],$$
which goes to show the difference in the dynamics of two distinct areas at a given time point, hence defining a “cross-difference-in-difference”. If, e.g., two structurally similar districts *i* and *j* both follow exactly the same non-pharmaceutic intervention schedule, Equation (7) should then yield a time series constantly close to zero.

To complete this, we use cross-correlation analyses based on Kendall’s correlation coefficient in order to quantify mutual associations of the dynamical patterns of areas (counties or federal states) expressed via the associated auto-difference-in-difference time series. Hierarchical clustering is applied in the “Ward.D2” mode.

## 3. Results

### 3.1. German SARS-CoV-2 Epidemic Activity Geographically Clusters into East and West

#### 3.1.1. Allowing for a Visual Exploration through Dimensionality Reduction

Dimensionality reduction of the 400 county-specific incidence time series leads to patterns within the two-dimensional target space, as depicted in Figure 1. Remarkably, three different commonly used reduction algorithms basically yield the same pattern: a clear separation into an Eastern and a Western German cluster can be observed. Panels A and C each show a decomposition into two principal components. The two panels differ only in the choice of subsets for which normal data ellipses have been calculated. Panel A shows normal data ellipses for the two subsets of counties that belong to West and East Germany, whereas panel C shows the ellipses for subsets corresponding to the 16 federal states separately. The locations of the ellipses belonging to the five Eastern German States can clearly be distinguished from the ellipses corresponding to Western German States. Although the clusters slightly overlap, the East–West dichotomy is clearly visible.

This result also applies to the right panels of Figure 1, which show t-sne (panel B) and UMAP (panel D) transformations, respectively. A few data points corresponding to Eastern German counties are located at the periphery of the Western German cluster; however, the two centres of mass are clearly separated.

#### 3.1.2. Canonical Correlation Analysis Provides Added Values to the Findings

The aforementioned results can be confirmed by applying a canonical correlation analysis. Correlation matrices corresponding to Pearson’s and Kendall’s correlation coefficients, respectively, are depicted in Figure 2. Also shown are two clusters for each correlation matrix resulting from Ward’s hierarchical cluster analysis. Application of the very same cluster algorithm does lead to a strict separation of Eastern and Western German States when being applied to Kendall’s correlation matrix (lower panel) in contrast to the application to Pearson’s correlation matrix (upper panel). Specifically, hierarchical cluster analysis following a linear correlation analysis leads to the allocation of the two West German states Bavaria (BY) and Baden-Wuerttemberg (BW) to the cluster otherwise dominated by East German states. As we learned from the dimensionality reduction above, points corresponding to counties belonging to BY and BW, respectively, are located at the interface between East and West German clusters (cf. Figure 1). Since the pairwise correlations of incidence time series cannot be expected to be strictly linear, the hierarchical clustering following Kendall’s correlation appears to be more convincing. It is compatible with what we learned from visual inspection of Figure 1.

#### 3.1.3. Consolidation of the Observed Clusters through Network Visualisation

Remarkably, the previous conclusions can also be confirmed in the form of network visualisations (Figure 3 and Figure 4) when being applied to incidence data aggregated to the federal state level. The network shown in Figure 3 results from an MDS based on Kendall’s correlation coefficients. Specifically, the edges between the vertices (federal states) represent partial (Kendall) correlations derived from graphical LASSO (correlation strength mapped to line width, positive correlations are coloured green, negative red, respectively). Spatial arrangement results from similarities also calculated from Kendall’s correlation coefficients, i.e., from the corresponding MDS. Once again, Eastern and Western German states are clearly separated into well-distinct clusters.

Basing the spatial arrangement on principal components instead of correlations yields the network structure depicted in Figure 4. The graph clearly tells us that one principal component would be sufficient to explain the variability of the time series. The nodes belonging to Western German states relatively tightly cluster at one end of this component, whereas the nodes representing the Eastern States extend over a greater length but are still well-separated from the Western German cluster. Thickness and colour of edges obey the same rules as in Figure 3. Since PCA is based on a reduction that optimises variability, it may be conceived as the most evident result, taking into account, however, that it results from the linear modelling constraint by the corresponding assumptions. In summary, the results from different approaches to dimensionality reduction are largely in agreement. Thus far, a striking difference in the epidemic dynamics of East and West Germany can safely be concluded.

To conclude this section, four commonly communicated centrality indicators are presented in Figure 5. Of note, these indicators are identical for the two networks presented in Figure 3 and Figure 4 since both networks are based on the same correlation matrix. All four indicators are similar, which is not unusual for the given case of connections of nodes, which are not strictly binary (connected vs not connected). The strength of a given node is simply the sum of the absolute values of all pairwise correlation coefficients of this node within the entire network. However, the X-axes of all four centrality indicators are transformed to z-scores rather than showing the raw values, which allows for a straightforward comparison. Obviously, the expected influence and strength do not make much of a difference. Slight differences between strength and expected influence, as can be observed, e.g., for NW, stem from a small number of low magnitude negative correlations considered when calculating expected influence. Assume that given a particular index node that sits between two nodes, which are not strongly correlated, then a strong betweenness entails strong correlations for the index node with respect to the neighboured nodes. As an example, Saxony-Anhalt (ST) does strongly correlate with both MV and TH, but without a substantial correlation between the latter two federal states (cf. Figure 3). This substantially increases the betweenness of ST. However, overall betweenness and strength do not differ substantially for most of the nodes. The same is true for closeness.

In summary, the indices may not contribute very much to the understanding of the prevailing dynamics; nevertheless, the reader should not be deprived of the information they provide—according to the motto, no result is also a result.

### 3.2. Variability of County-Specific Fold Changes in Reproduction Numbers Correlates with Spatial Heterogeneity

SARS-CoV-2 spatio-temporal homogeneity in Germany is depicted in different facets in Figure 6. To start with, age-independent incidence time series of all 400 German rural/urban districts (counties) (Equation (1) aggregated over all age classes) by and large follow the same wave-like shape as observed for the pan-German incidence curve (Equation (1) aggregated over all age classes and over all counties), however, exhibiting rather wide variations in magnitude (see Figure 6A). Even on the level of the 16 federal states, the individual curves deviate considerably from each other. The ranges depicted in Figure 6A give a vivid expression.

The variations of county-specific incidence curves are the result of continual dynamical changes expressed by a dense series of spikes of weekly fold changes in the instantaneous effective reproduction numbers as calculated from Equation (5) on the pan-German, as well as on the county and federal state levels, respectively, as shown in Figure 6B. Hardly surprising, these differences in oscillatory patterns in the time domain show up as spatial heterogeneity expressed via the diversity measure calculated from Equation (3), as shown in Figure 6C. A non-constant homogeneity over time points to a residual coherence between the auto-difference-in-difference time courses. Thereby, spatial homogeneity (diversity) has been calculated and is presented both with and without stratification by three age classes (i.e., kids (age < 18 years), adults (18 years ≤age< 60 years), seniors (age≥ 60 years), all ages).

As expected, at the outset of the German COVID-19 epidemic for all age classes, spatial diversity starts at a value close to zero, followed by a rather steep increase roughly within the first 4 to 6 epidemic weeks (Figure 6C). For all three age classes, the diversity curves show relatively sharp and short-lived dips that begin shortly before the respective holiday periods of the three observation years. The slump was particularly pronounced in 2020, only slightly smaller in 2021 and more moderate in 2022. These dips, at least for 2020 and 2021, coincide with low incidence periods (Figure 6A) but also with periods of high relative changes in reproduction numbers (Figure 6B). This behaviour also holds for some smaller intermittent drops in diversity.

Indeed, the logarithm of the age-independent SARS-CoV-2 incidence time course (Figure 6A) strongly correlated with age-independent diversity (Figure 6C) yielding Kendall coefficient 0.74 (p<0.001). Likewise, the Kendall coefficient of correlation between the time course of the logarithmised range of the age-independent auto-difference-in-difference taken over the counties (Figure 6B) and age-independent diversity (Figure 6C) amounts to −0.6 with p<0.001. Thereby, the first 9 weeks were removed from the time series due to unreliable diversity estimates at the outset of the epidemic. The activity in 2022 is somewhat out of line overall, although the indicated behaviour continues at least moderately.

### 3.3. Spatial Homogeneity of Child Incidence but Increased Overall Heterogeneity in the East

Remarkably, the diversity curve corresponding to the youngest age class (children and juveniles) remained on top of the two other curves, which corresponded to the adults and seniors during the course of time until the curves apparently converge towards the end of 2021. From roughly April 2022 onward, diversity corresponding to the adult age class started again to drop and, therefore, diverged from the two other curves, which remained in-phase at almost identical magnitudes.

If we calculate the diversities for West and East separately, we obtain the same ordering pattern by age group (not explicitly shown). However, the diversity curves for West Germany were larger in magnitude than those for East Germany over almost the entire time course. The difference of diversity between West and East is depicted in Figure 6D, which remained positive most of the time. To summarise the findings so far, increased variability of county-specific fold changes in reproduction numbers correlated with increased spatial heterogeneity and coincided with a drop in incidence. East German COVID-19 incidence exhibited a considerably stronger spatial heterogeneity than observed for the West.

### 3.4. Decreasing Trend in Fold Changes in Reproduction Numbers

A closer look at the courses of age-independent federal state-specific fold changes in reproduction numbers calculated according to Equation (5) reveals a clear overall trend in decreasing magnitudes (Figure 7). Although later “epidemic waves” have much more pronounced magnitudes than early “waves” (cf. Figure 6A), their rates of change appear to be more moderate. The 16 federal state-specific auto-difference-in-difference curves are depicted in Figure 7A along with a 5-week-windowed envelope. More concrete, at each point in time, *t*, the envelope showed maximum and minimum values within the window time [t−5,t+5] (in weeks). The evolution of density curves as shown in Figure 7B further clarifies this converging behaviour. The gradual narrowing of the densities, calculated per quarter, is striking.

The district of Heinsberg, located in North Rhein-Westfalia (NW), is known for the first “super-spreading” event, which arguably sparked the COVID-19 epidemic in Germany (cf. [44]). Unsurprisingly, after a quick rise of numbers of infected individuals, the first attempts to mitigate the epidemic by means of lockdown orders took effect in NW, which at least partially explains the strong acceleration and deceleration during the first weeks of the epidemic. A similar pattern was observed for the second smallest federal state Saarland (SL) and the second most populous federal state Bavaria (BY). The latter state is known to have seen the first SARS-CoV-2 index case (cf. [45]), although without super-spreading event. Some states, particularly East German states as MV, ST, TH, BB, SN (cf. the list of abbreviations), show moderate de-/accelerations during the first wave but more pronounced changes in reproduction numbers during the second wave. In Baden-Wuerttemberg (BW) the amplitude of fold changes in reproduction number remained strikingly low and approximately constant during the entire epidemic. Apparently, social conduct did not change considerably in BW during the epidemic, although this is speculation.

To summarise, a common trend in the long run over all German federal states of decreasing amplitudes of fold changes in reproduction numbers can be observed; however, there are state-specific differences with respect to intermittent bursts.

### 3.5. Pronounced Fold Changes in Reproduction Numbers for the Younger and the Elder Cohorts

Throwing a glance onto age-stratified time courses of auto-difference-in-difference reveals striking age-dependent differences. Figure 8 depicts the auto-difference-in-difference curves corresponding to five arbitrarily chosen West German federal states separately for the 10 age classes. Likewise, the age-specific auto-difference-in-difference curves for the five East German federal states (Berlin excluded) are shown in Figure 9. The younger cohorts up to age 18 y and the elder from age 60 y upwards unveil strong amplitudes of fold changes in reproduction numbers whereas the corresponding amplitudes of the adult (medium aged) cohorts remain moderate throughout the epidemic. During the epidemic, enormous efforts were made to protect the elderly. It is therefore easy to understand that shelter-in-place or isolation orders showed the greatest effect for the seniors compared to the employed people. At the same time, the political controversies led to an inconsistent and erratically changing set of rules and regulations. This is just as true for the youngest cohort, the children and juveniles. School closing orders have been replaced by school opening orders in a discordant and somewhat haphazard fashion—a behaviour which has also been called “flying blind” [46]. While the impact of these discordant rules on the epidemic activity is of course speculative, it can safely be stated up to this point that the de-/acceleration of this activity, i.e., the fold changes in reproduction speed, is by far more pronounced for both the young and the elder subpopulation, but not so for the medium-aged adult cohort. In exactly this sense, kids and juveniles, as well as seniors, are the driving factors of the epidemic, at least for the West German federal states. For it is the case that a striking difference between the West and the East German states can be observed. The differences between the medium-aged and the other (junior and senior) cohorts was less strong or even absent for the East German states. Within the West German set of states, Schleswig-Holstein (SH) is an exception in that the oscillation of auto-difference-in-difference resembles the corresponding East German patterns.

In summary, both the younger (up to 18 y) and the elder (60${}^{+}$ y) cohorts show stronger changes in SARS-CoV-2 reproduction numbers when being compared to the medium-aged adult subpopulation. In this sense, kids, juveniles, and seniors drive the epidemic stronger than working adults. We hypothesise that containment and isolation measures are less actionable for working people. It is perhaps more difficult to explain the East German dynamic patterns, which appear to be much more similar with respect to age classes and, at the same time, show much more pronounced amplitudes for the fold changes of the reproduction number. This finding is compatible with our results above-derived from cluster analyses. Our previous findings in [12] draw us to the conclusion that the well-known political and socio-structural differences between East and West Germany are proper surrogates for the underlying mechanisms.

### 3.6. Federal States Exhibit Dynamic Dissimilarities

As shown in the previous sections, the rates of change of the reproduction numbers calculated per age class and per federal state in the course of time do produce age-dependent patterns but appear also to exhibit state-dependent dynamical features. In this section, a closer look is taken at the differences and similarities resulting from state-by-state comparisons. Equation (7) is considered an appropriate time-dependent measure that captures mutual dynamic similarity. Indeed, Equation (7) can be conceived as a direct application of the original difference-in-difference concept. The logarithm of the ratio defined in Equation (7) is expected to yield constantly zero if the dynamical features of the two states under comparison are identical. Figure 10 depicts all pairwise comparisons with reference state North Rhine-Westfalia (NW). Obviously, all 15 comparator states have a markedly different dynamical patterns at the outset (roughly the first 3–6 months) of the epidemic, with the most extreme differences observed for SL and BY. For some of the comparator states, particularly HE, NI, and RP, the pronounced initial oscillation fades out to a very small amplitude. A look at Figure 2 reveals that the Kendall correlations of the incidence time series also go in the same direction, namely slightly smaller coefficients for the NW-SL and NW-BY correlations, compared to the other three pair correlations mentioned above. We hypothesise that the corresponding federal states exhibit a more coherent dynamical pattern. The corresponding curves for each of the remaining 15 reference states are shown in Appendix A.

For a more condensed or integral comparison, it is hypothesised that Kendall’s correlation coefficient is another valid quantification that captures dynamical similarity. The resulting correlation plot is shown in Figure 11. The correlation plot is ordered in hierarchical clustering mode using agglomeration method “Ward.D2” as before, constraint by two clusters. Changing the agglomeration method or the chosen number of clusters leads to a considerable fluctuation of clustering patterns (not shown). It turns out that the canonical pairwise correlations of auto-difference-in-difference curves are comparably less sensitive then the method defined by Equation (7) to detect shared dynamical patterns of the federal states via hierarchical clustering.

## 4. Discussion

The presented semi-quantitative and broadly descriptive way of proceeding to assess age- and county-specific COVID-19 incidence time series recorded in Germany revealed informative patterns. Most striking is the fact that the time series can be allocated to two geographic similarity classes or clusters which coincide with the geopolitical division into East and West Germany. Spatial homogeneity of COVID-19 incidence is waxing and waning in the course of time for each age class, however, remains highest for the youngest age class (children, adolescents) during the entire time course.

Temporal acceleration patterns do differ by geography and age. Generally, for all federal states, the dynamical changes in terms of fold changes of reproduction number gradually fade out in the course of time, although some intermittent bursts can be observed. Particularly striking is the fact that the dynamics for Baden-Wuerttemberg (BW) are relatively flat over the entire observation period. In sharp contrast, Saarland (SL), a very small federal state, does show the largest variability in fold-changes of the reproduction number. The cross-difference-in-difference method applied to all pairs of state-dependent incidence time series reveals patters of similarities and dissimilarities that have to be discussed from the perspective of geopolitical differences, which is beyond the scope of this contribution. However, we point to the observation that North Rhine-Westphalia (NW) and Hesse (HE) do not only exhibit similarity both in terms of cross-difference-in-difference and Kendall correlation, but these two federal states are also located in the center of the Western German cluster derived from a network analysis (Figure 3). In this sense, NW and HE are “average states” with respect to the SARS-CoV-2/COVID-19 epidemic. The reader is encouraged to throw a glance on the series of figures in Appendix A. Among many other interesting patterns, Saarland (SL) exclusively shows very pronounced mutual cross-difference-in-difference curves and, therefore, makes SL unique in a certain sense.

In West Germany, children and juveniles as well as seniors do contribute more intensively to de-/acceleration of the epidemic spread when being compared to the middle-aged class. Such a difference cannot be observed in East Germany where all age classes equally strongly contribute to fold-changes of the reproduction number.

A proper interpretation of the results obtained requires a discussion of our approach. Only a couple of months after the pandemic outbreak of SARS-CoV-2/COVID-19, RJ Klement presented a systemic picture, including close to one hundred components to build complex functional relations [47]. Although Klement’s network of “causal” relations is of rather wide scope and included interactions within and between levels of organisation (i.e., macro-micro interactions), it is still preliminary and far from exhaustive, as the author confirms. Klement thus contributed to a hermeneutic discourse on the nature of the pandemic, i.e., to its better understanding, but the question remains how such a complex picture can be operationalised. In practice, science is thrown back on ambiguous reductionist views of a few interacting components. Unprecedentedly, the conception of countermeasures based on both political as well as scientific criteria, indeed the entire culture of communication, has been overshadowed by an intra-scientific scramble for interpretive sovereignty as a consequence of this ambiguity [48]. This completely derailed communication culture undoubtedly itself added quite considerably to the mechanisms of the pandemic: an infodemic [7]. At least this important cluster of nodes should be added to Klement’s “causal” network.

“Systems thinking” is a collective term for very different conceptions of complex self-(dis)organised systems [49], even if, strictly speaking, the term self-organisation was only used extensively in the context of synergetics [50]. The scientific nature of systems thinking has been questioned and it has, apparently, been conceived as a hermeneutic process [51,52], thus belonging to the context of discovery rather than to the context of justification in terms of Reichenbach’s partition of the epistemic process [53]. The pressing need to better understand the COVID-19 pandemic, and the sheer lack of convincing and actionable evaluations, encouraged us to place greater emphasis on the benefits of systems thinking. It is due to the very nature of complex systems that no unique optimal tool exists for their analysis. Each analysis tool allows to take a particular look at the system and the combination of such analyses reveals the different facets of the given complexity.

The strategy followed here can be described as an attempt to largely dispense with model assumptions and simply let the data speak for themselves. In this sense, the approach is non-parametric and largely descriptive. The limitation of this approach is at the same time its strength. No causal relationships, or even functional dependencies, are shown. An important note on this: causality is an a priori, i.e., a fundamental principle that cannot be derived from empiricism—not even from experimental empiricism. At best, a randomised controlled trial (RCT) provides evidence for functional dependencies under very specific conditions. Less evidence is attributed to results of regressions in the context of observational studies, even if the result thus obtained is highly self-evident. But be that as it may, at the end of the day, the conclusion and, in our opinion, already the choice of study design, including RCTs, is always already value-based (cf. [54,55]). In the light of the above, the following interpretations of the results obtained should be deprived of dogmatic rigidity.

The observation that epidemic activity corresponds to the geopolitical East–West division of Germany confirms earlier findings based on regression analysis [12]. The dependence of this clustering on the chosen measure of similarity and reduction algorithm could give rise to a possible critique. However, this criticism can be countered by the fact that four different methods produce the same results.

The interpretation that the greatest spatial homogeneity of incidence for the youngest age group results from a spatially homogeneous background incidence of this age group may be going a bit too far, but a tendency in this direction is indicated. This view is supported by findings of increased seroprevalence among school-aged children (cf., e.g., [56,57,58,59,60], and references therein) as well as findings that schools are not inherently low risk [61,62]. Note, however, that a comparably lower virus load in children may lead to underdetection, which in turn, might also bias the calculation of spatial diversity. Furthermore, in East Germany we find a considerably larger spatial heterogeneity when being compared to West Germany. Seen in the light of marked geopolitical differences between East and West (cf. [11]), this result may appear less puzzling. In addition, it is in agreement with other spatio-temporal characteristics as discussed in the following.

The pronounced temporal changes of infection dynamics for children and adolescents is in agreement with what has been found in [24] using a different analytical approach which gives rise to the interpretation that children have a significantly greater influence as a driver of the pandemic than previously suspected. A possible explanation is the relatively unsteady dynamics of introduction and withdrawal of containment measures for kids, particularly in schools and daycare facilities. The pronounced impact of adults in the East on dynamical changes when being compared with the West German population may be due to carelessness or differing socio-political attitudes. Indeed, evidence is mounting on how influential sociocultural aspects and personal beliefs are in relation to epidemic activity. For a more profound discussion of this issue cf. [11,12,63,64,65,66,67].

Thus, with due caution we conclude that varying containment measures and their compliance, as well as regular occurrences such as school vacations are much more instrumental to change the behaviour of non-adult people relevant for the control of the epidemic reproduction number. We hypothesise, however, that the effects of school openings are strongly confounded by the current local epidemic conditions, i.e., the current effective reproduction numbers (in line with findings in [68]), and, even more important, by prevailing preparedness, facilities and reliance at the schools and accountable local authorities. Thus, on the one hand, school opening can contribute to combat the epidemic in case of a quick detection of an infection and a proximate shelter-in-place order. On the other hand, school opening can worsen the situation in case of overwhelming infectivity and concurrent lack of preparedness. The pronounced irregularity observed for the epidemic dynamics corresponding to the young population teaches to shift the focus regarding control measures toward children and adolescents. Further research is needed to better understand the causes behind the observed irregularities.

Whether the observed local differences can be attributed to real differences in incidences or rather to locally different frequencies of testing, i.e., different numbers of undetected cases, is unclear (cf. [69,70]). This is an unavoidable and probably the strongest limitation of evaluations that refer to publicly available registered case numbers. The observed spatio-temporal intermittency, i.e., the irregular alternation of phases, are presumably to some extent caused by the time-dependent detection ratio. Moreover, the existence of a depensation effect leading to a “detection threshold” cannot be ruled out. In other words, during a low incidence period, disposition to test might be particularly low, which might in turn entail unrealistically many “zero events”. To put this limitation in a slightly better light, we point out that the interpretations given here are only suggestions anyway. The observed dynamic patterns are objective, but the associated potential explanations, including changing vaccination rates and the like, are not at all.

Finally, the lack of reliable data on all types of contact regulations does definitely limit the explanatory power of our analysis. However, we are able to report intrinsic spatio-temporal patterns of the epidemic that now can be linked to all types of socio-cultural occurrences that are suspected to influence the transmission dynamics.

## 5. Conclusions

We presented the results from an explorative study and want to conclude by highlighting the captivating advantage of such a study: we let the data speak and did not use contestable model assumptions. Broadly speaking, the work provides a kind of atlas for spatio-temporal patterns of the epidemic, which now need to be interpreted with expertise from different disciplines. We encourage readers to combine the results presented here with those in [23,24] to obtain an even more comprehensive picture on COVID-19 pandemic dynamics. In particular, we urge sociologists and policy makers to associate the observed processes of change with both sociocultural characteristics of individual regions and local policy-making processes. The approach in the presented study was essentially inductive and therefore requires verification/falsification of the hypotheses put forward on the one hand, but is less prone to dogmatic bias on the other. Finally, we hope that our study can also make a methodological contribution to getting a handle on the unusual complexity of the COVID-19 pandemic.

## Figures and Tables

**Figure 1 entropy-25-01137-f001:**
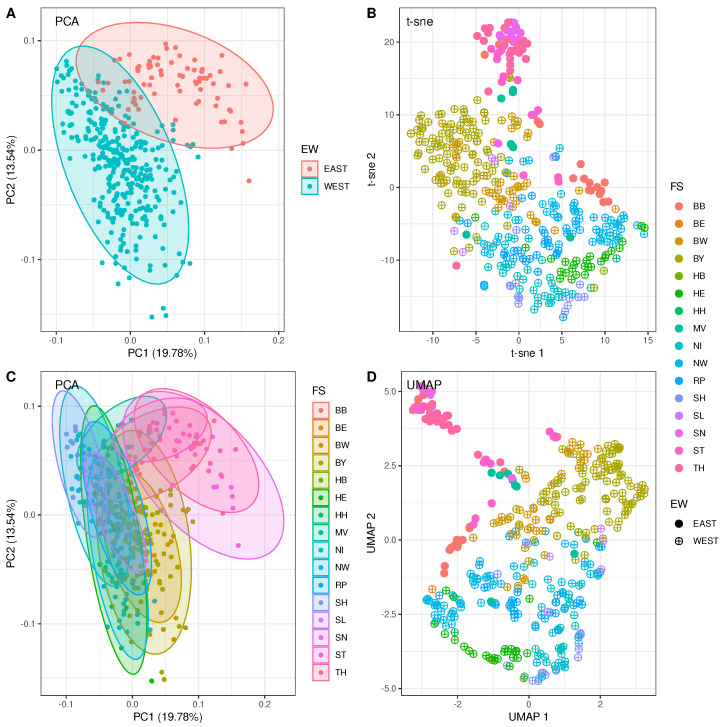
County-specific total incidence time series after reduction to two dimensions. (**A**,**C**) Principle component analysis (PCA) along with normal data ellipses embracing East and West Germany (**A**) and the 16 federal states (**C**), respectively. X- and Y-axes labels contain percentages of explained variability by the corresponding component. (**B**) Dimensionality reduction using t-sne with perplexity=30. Full circles point to incidence time series observed in East German counties, whereas circled crosses refer to West German counties. (**D**) Dimensionality reduction using UMAP with *n*-neighbors=10. Usage of markers as in (**B**).

**Figure 2 entropy-25-01137-f002:**
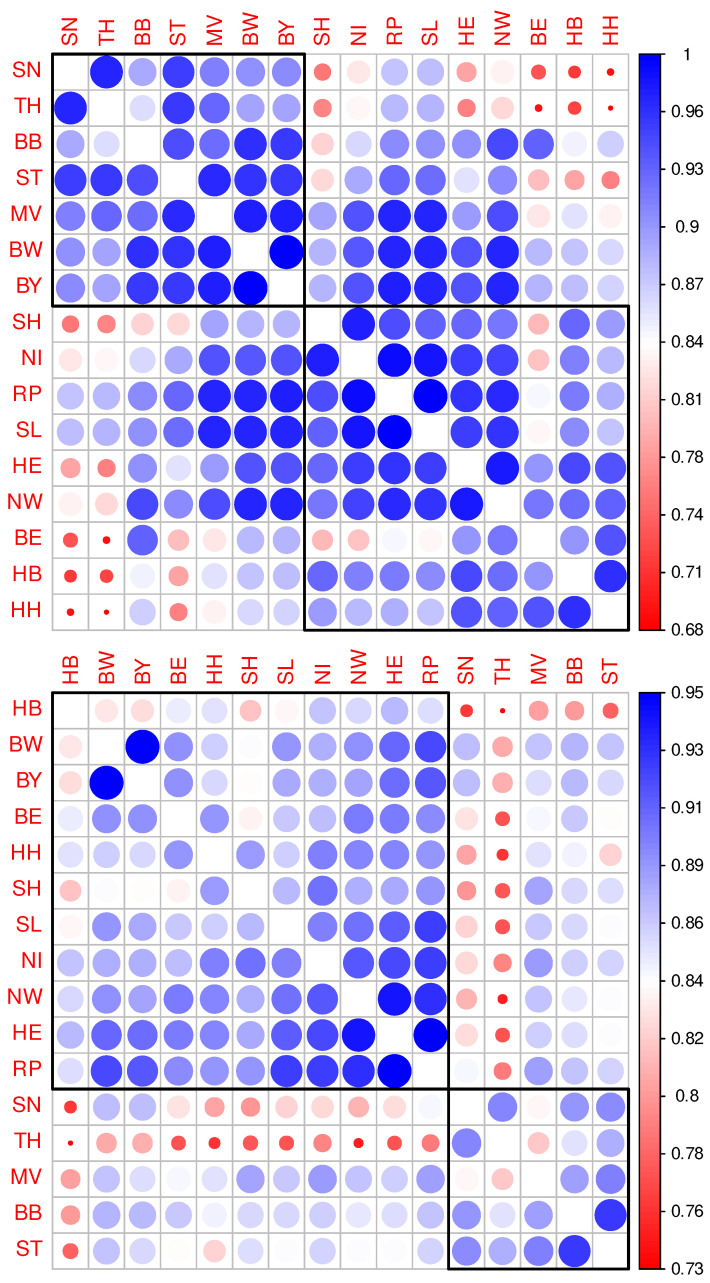
Correlation matrix corresponding to the incidence time series with hierarchical clustering. Upper panel: Heatmap-like depiction of correlation coefficients resulting from pairwise Pearson correlations of federal state-specific incidence time series along with hierarchical clustering restricted to two clusters. Lower panel: Correlation matrix as in the upper panel, however, calculated on the basis of the Kendall’s correlation coefficients.

**Figure 3 entropy-25-01137-f003:**
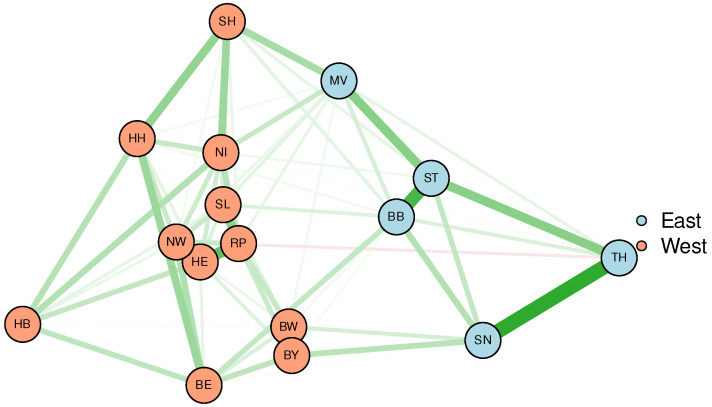
Network visualisation of the federal state-specific incidence time series based on MDS. For the spatial arrangement, similarities are calculated from Kendall’s correlation coefficients. Colour (green corresponds to positive and red to negative correlations, respectively) and strengths of edges are likewise derived from these coefficients.

**Figure 4 entropy-25-01137-f004:**
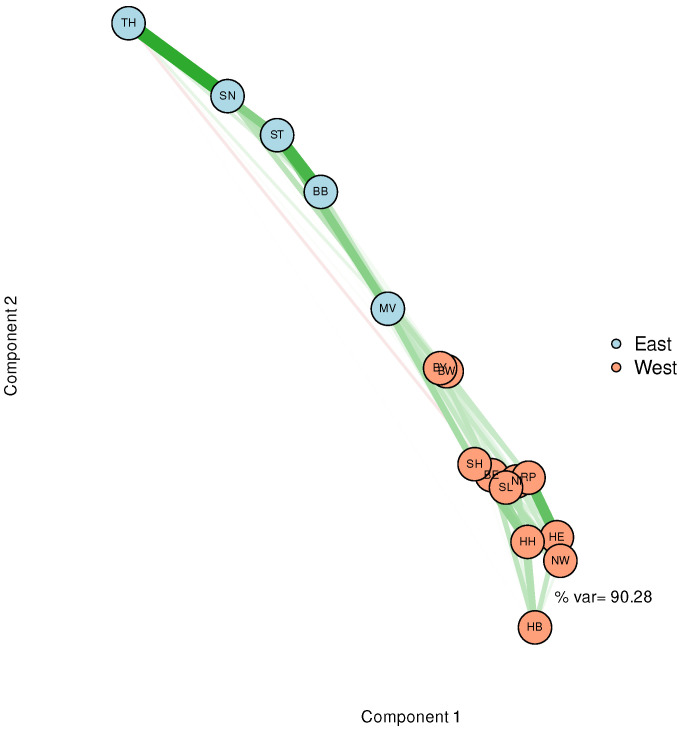
Network visualisation of the federal state-specific incidence time series based on MDS. For the spatial arrangement, similarities are calculated from a PCA. Colour (green corresponds to positive and red to negative correlations, respectively) and strengths of edges are likewise derived from these coefficients.

**Figure 5 entropy-25-01137-f005:**
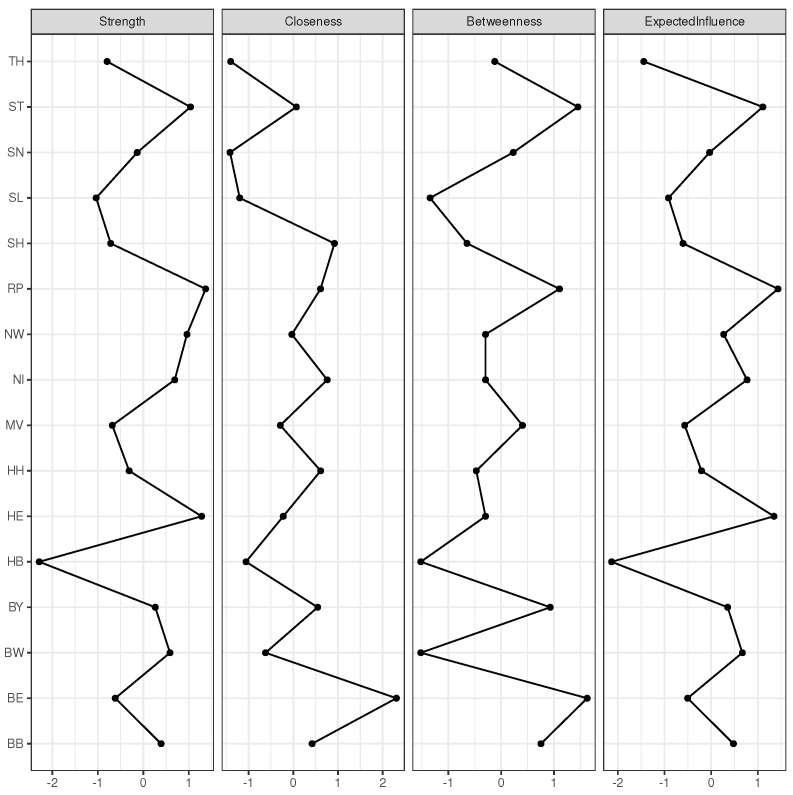
Centrality indicators of the network depicted in Figure 3. X-axes are scaled as z-score.

**Figure 6 entropy-25-01137-f006:**
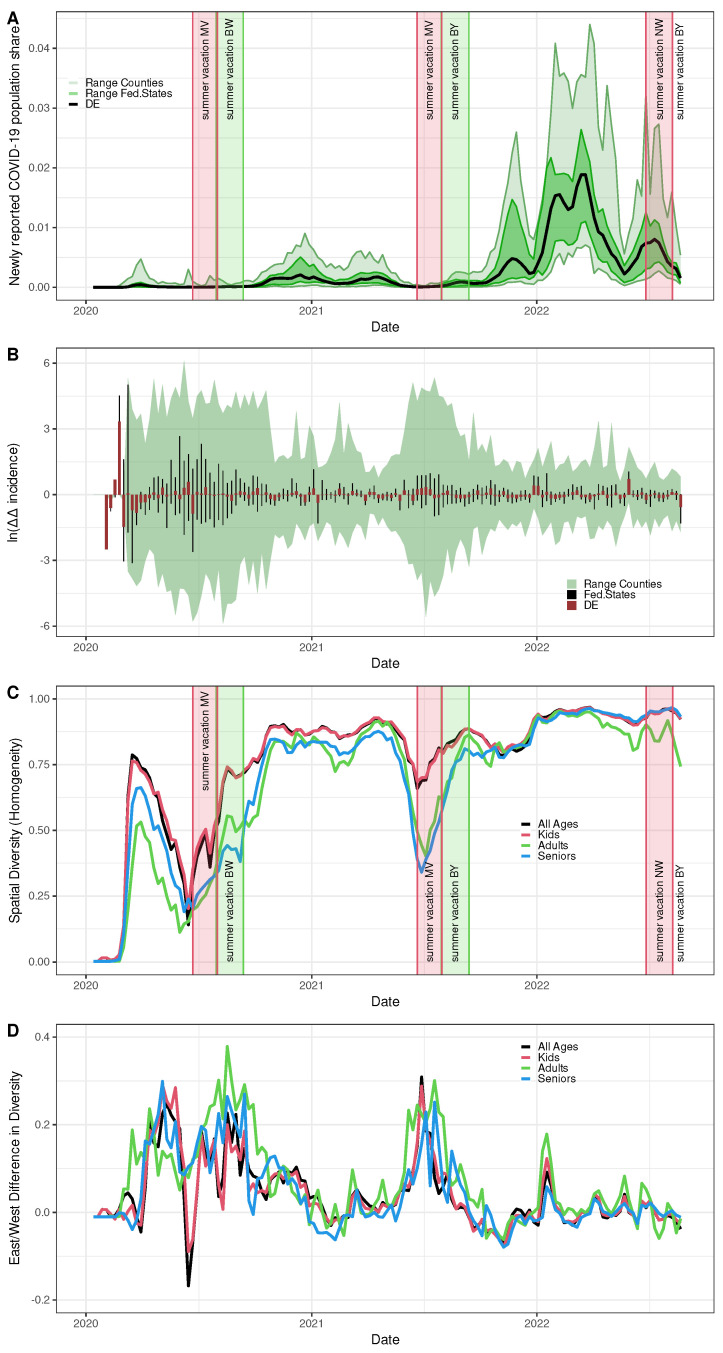
Spatio -temporal patterns of SARS-CoV-2 incidence. (**A**) Pan-German incidence, i.e., weekly new cases by population size (black line), range spanned by state-specific incidence curves (dark green area), and range spanned by the county-specific incidence curves (light green area). (**B**) Auto-difference-in-difference time series for the pan-German incidence (red bars), the 16 federal states (black needles, slightly displaced for better visibility), and the 400 German counties (maximum to minimum range). (**C**) Age-stratified time courses of the spatial homogeneity of incidences over 400 German counties given by Shannon’s diversity measure. (**D**) West-East difference of diversity. (**A**,**C**) Red and green rectangles show the first (MV or NW, resp.) and the last (BY or BW, resp.) summer vacation in 2020, 2021, and 2022, respectively.

**Figure 7 entropy-25-01137-f007:**
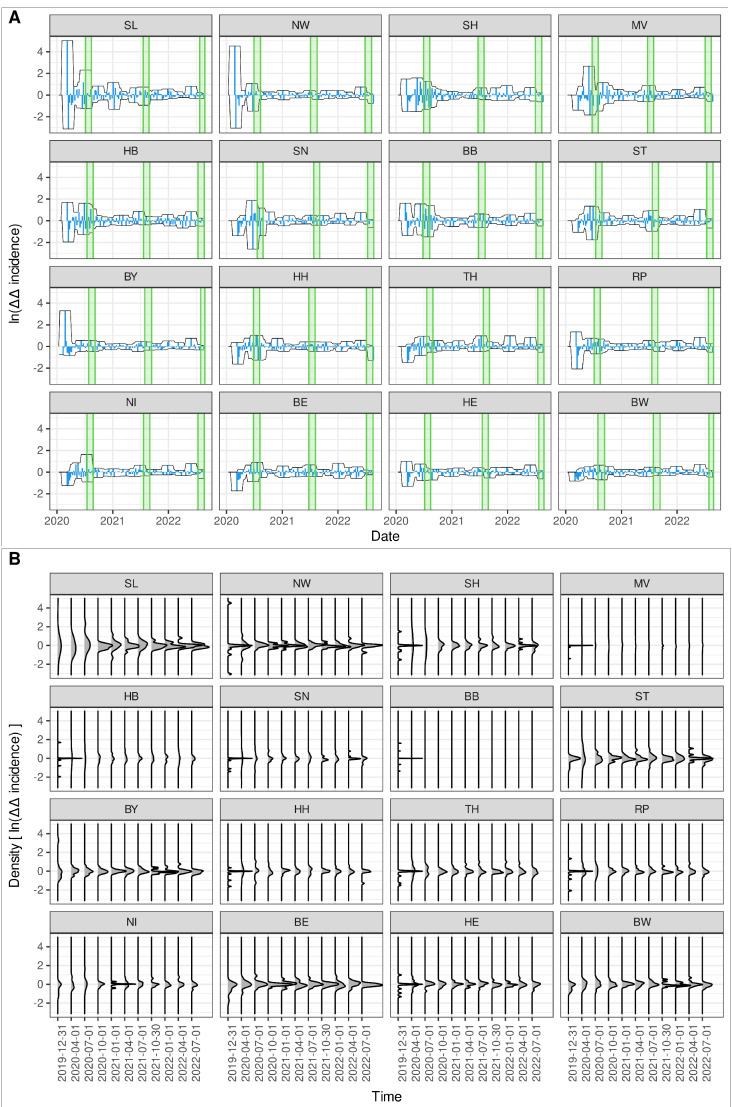
Auto-difference-in-difference per federal state. (**A**) Time series of the auto-difference-in-difference according to Equation (5) calculated per federal state (age- and sex-aggregated) along with the corresponding summer vacations periods in 2020, 2021, and 2022 (green shaded areas) in descending order of variance. The curves are enclosed by a 5-week-windowed envelope for visualising trends. (**B**) Evolution of corresponding density functions calculated per quarter for each auto-difference-in-difference curve.

**Figure 8 entropy-25-01137-f008:**
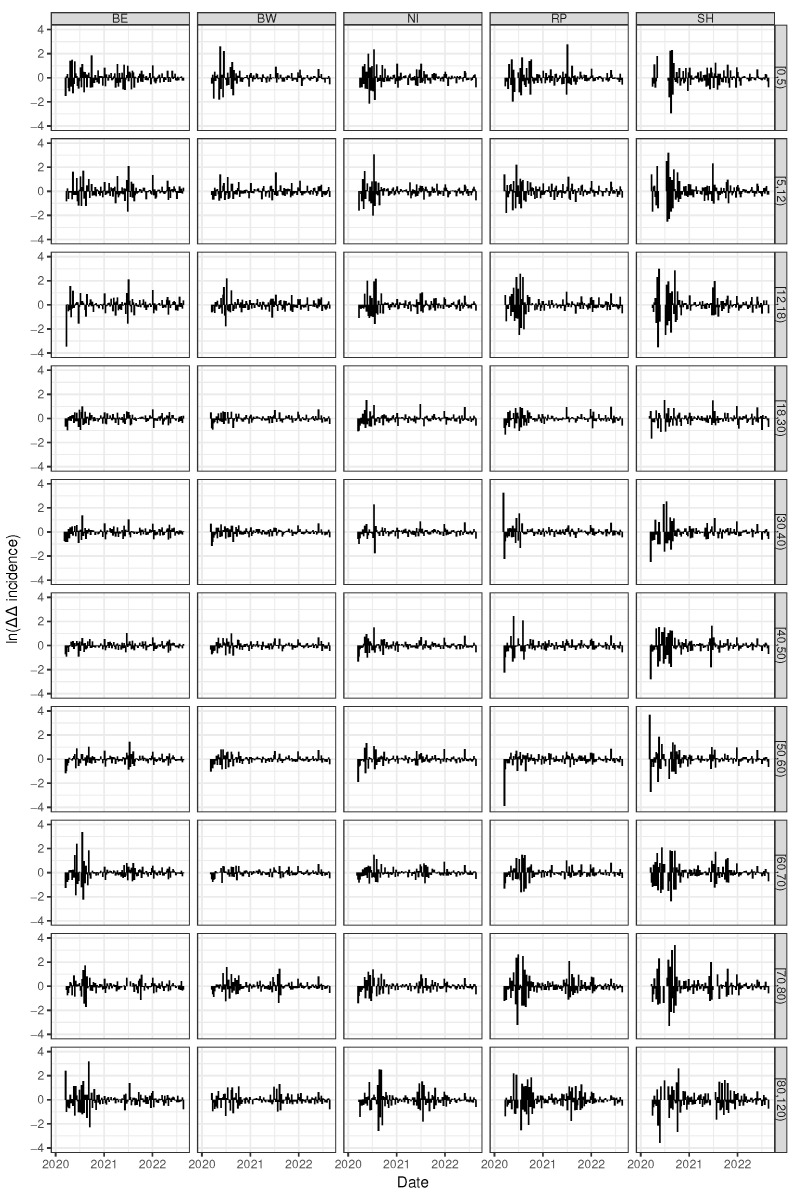
Time series of age-dependent (age classes ordered from top to bottom) auto-difference-in-difference curves according to Equation (7) shown for 5 arbitrarily chosen Western German federal states (horizontally arranged).

**Figure 9 entropy-25-01137-f009:**
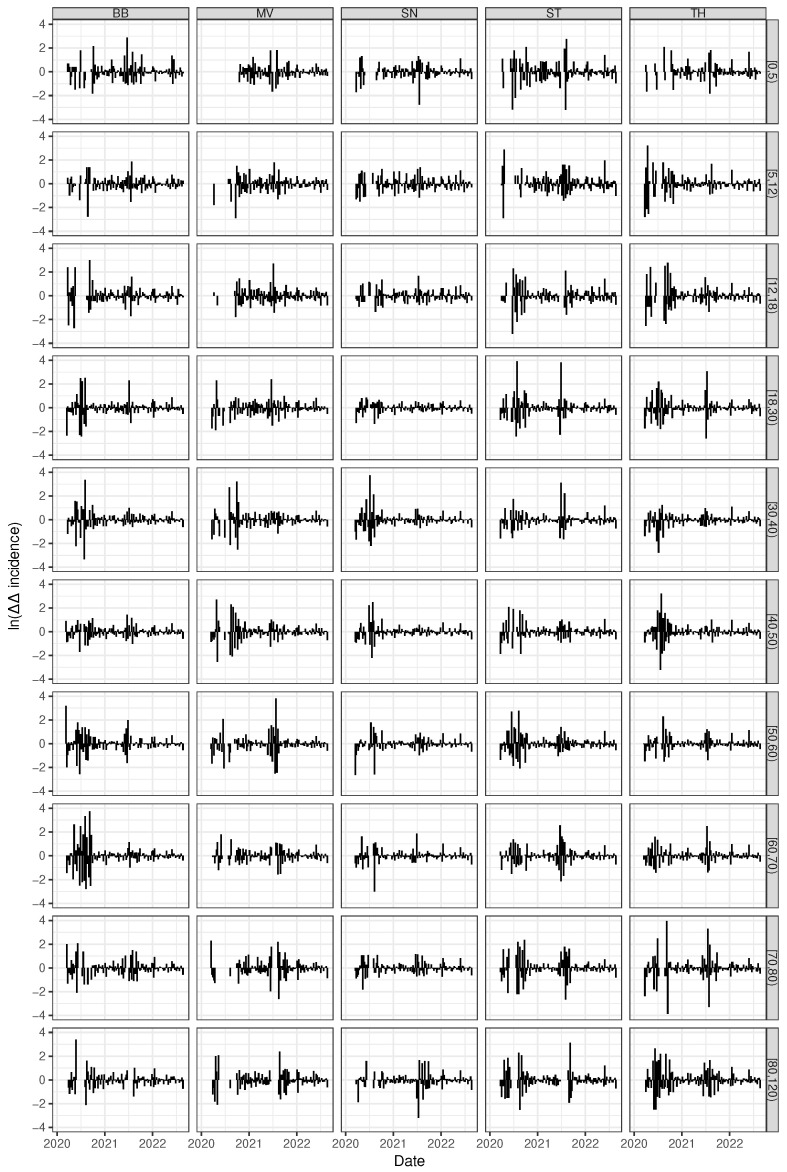
Time series of age-dependent (age classes vertically ordered) auto-difference-in-difference curves according to Equation (7) shown for the 5 Eastern German (excl. Berlin) federal states (horizontally arranged).

**Figure 10 entropy-25-01137-f010:**
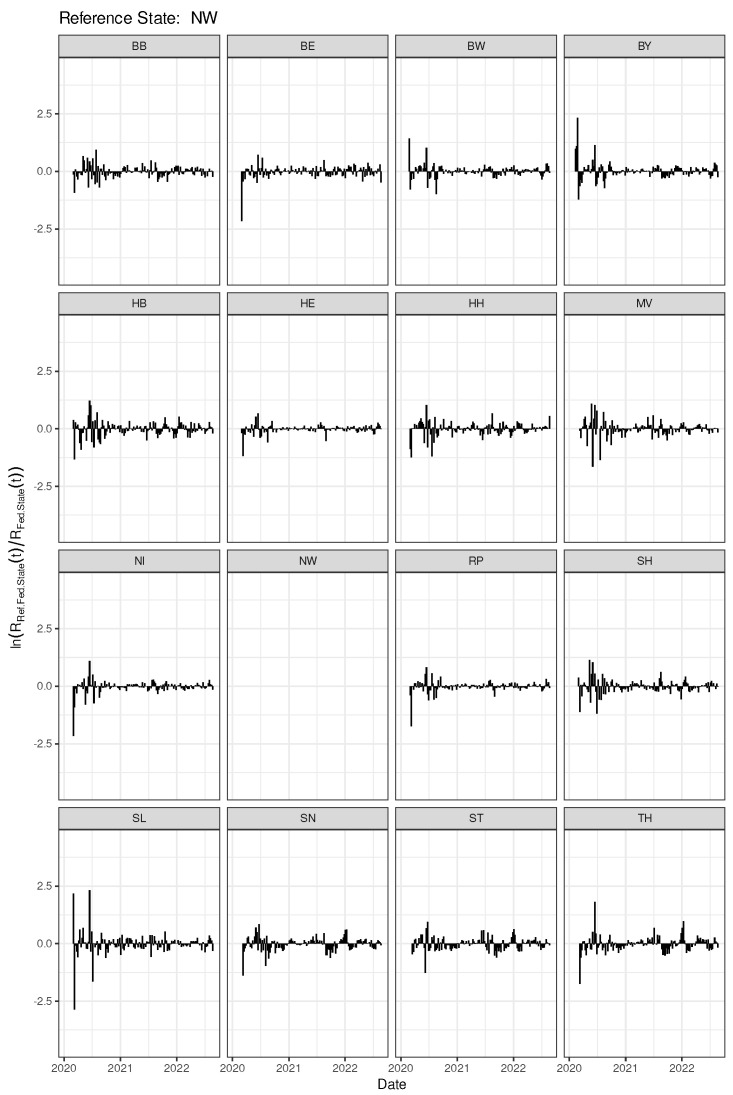
Time series of the cross-difference-in-difference according to Equation (7) calculated for NW versus all 16 federal states. The empty NW-panel is kept to easily spot NW as the reference federal state.

**Figure 11 entropy-25-01137-f011:**
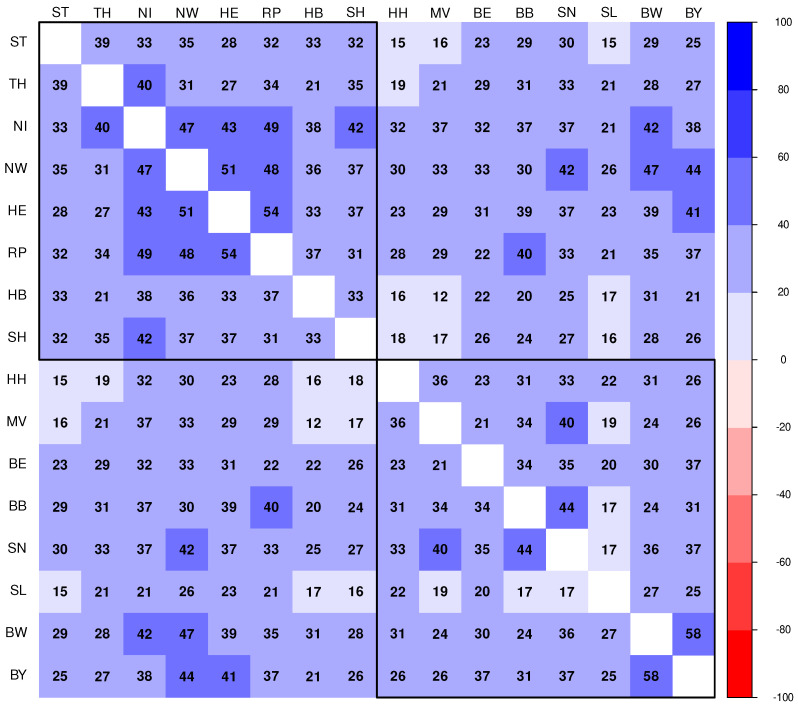
Correlation matrix showing coefficients (in %) of pairwise Kendall’s cross-correlations of auto-difference-in-difference time series $\ln\left(\Delta \Delta I_{i}\left(t\right)\right)$ (Equation (5)) corresponding to the pair of federal states as indicated by the row and column labels, depicted in hierarchical (Ward.D2) clustering mode.

## Data Availability

Publicly available data provided by [26,27] have been used exclusively.

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
