# Peer review of "Spatio-Temporal Patterns of the SARS-CoV-2 Epidemic in Germany"

_entropy, 2023, doi:10.3390/e25081137_

Round 1

Reviewer 1 Report

I am overall with a good impression in regard to the style of presentation and clarity of the ideas, I strongly feel we need more opinions like this one, born from natural observation of the phenomenon.

Introduction

Lines 20-37 offer a personal interpretation, which we lack many times, lately. I appreciate the point of view.

However, lines 38-52 should be included in the research method, representing the proposed direction and not an already explored basis, from which the research starts. The introduction needs more support with references from the literature.

Lines 52-63 should be moved to discussion, and lines 64-72 to methods.

I suggest redoing the introduction, with organization on 3 pillars: why the approach is important, what it brings new and where it starts from.

Methods: The study methodology must be described, before entering the actual description, which, anyway, I suggest to be compressed, maybe even in a language accessible to the general public, which is less interested in the working formula, more interested in what is achieved through the respective methods, as a predictive model.

Discussions

I prefer that this chapter be preceded at the beginning by an overview of the results obtained, with the appreciation of the personal contribution, of the specificity of the study; regarding the limitations of the study, they were well described.

The conclusions: it should emphasize what the study brought new and more perspectives for the future; it is no longer necessary to contain what has already been mentioned.

I suggest increasing the number of references.

MInor revision of the language, but still neeeded.

Author Response

RESPONSE TO REVIEWER 1:

REVIEWER 1:

Introduction

Lines 20-37 offer a personal interpretation, which we lack many times, lately. I appreciate the point of view.

ANSWER:
Thanks a lot indeed for your generous remark.

REVIEWER 1:

However, lines 38-52 should be included in the research method, representing the proposed direction and not an already explored basis, from which the research starts. The introduction needs more support with references from the literature.

Lines 52-63 should be moved to discussion, and lines 64-72 to methods.

ANSWER:
I followed the Reviewers recommendations to re-order the manuscript. All paragraphs have been moved to the proper sections with a few adaptations to maintain a smooth text flow.

REVIEWER 1:

I suggest redoing the introduction, with organization on 3 pillars: why the approach is important, what it brings new and where it starts from.

ANSWER:
The intro has been re-organized following the Reviewer’s suggestions.

REVIEWER 1:

Methods: The study methodology must be described, before entering the actual description, which, anyway, I suggest to be compressed, maybe even in a language accessible to the general public, which is less interested in the working formula, more interested in what is achieved through the respective methods, as a predictive model.

ANSWER:
Lines 38-52 from the introduction have been moved to the methods section and slightly adapted to meet the Reviewer’s recommendation. In think the description of the methods used should now be readable by both the general reader and the relevant expert. The idea behind the formulas should also be comprehensible, even if one skips the mathematical passages.

REVIEWER 1:

Discussions

I prefer that this chapter be preceded at the beginning by an overview of the results obtained, with the appreciation of the personal contribution, of the specificity of the study; regarding the limitations of the study, they were well described.

ANSWER:
I followed the Reviewer’s recommendation and re-structured the discussion accordingly. The discussion now starts with a brief summary of the results and then enters the actual discussion which has been adapted according the suggestions.

REVIEWER 1:

The conclusions: it should emphasize what the study brought new and more perspectives for the future; it is no longer necessary to contain what has already been mentioned.

ANSWER:
Unnecessary repetitions have been deleted and I put more emphasis on novelty and perspectives.

REVIEWER 1:

I suggest increasing the number of references.

ANSWER:
The number of references has substantially been increased both for the introduction and the discussion sections.

I would like to thank Reviewer 1 for their constructive and very helpful comments and recommendations. Particularly the suggested re-structuring of the sections substantially improved readability as well as the course of argumentation. I also agree that the manuscript was lacking supporting references. Some typos have been corrected and I went through the grammar.
Thanks indeed for that.

Reviewer 2 Report

Overall, I think this manuscript is methodologically sound, and represent a meaningful addition to the literature. However, there are a couple of issues that I would like raise, which are detailed as below:

1. For Equation (3), I believe it is a measure of homogeneity or similarity (ie, the opposite of heterogeneity), because the larger the D(a,t), the less spatial heterogeneity. (When entropy reaches minimum [ie, equal incidence over all locations], D(a,t) reaches maximum, which is 1, right?)

2. Following comment #1, Line 184-185, based on my understanding, should be “equal incidence overall locations gives minimum entropy, hence maximum homogeneity D(a,t)=1”. Or could you please explain if I misunderstood anything?

3. Lines 361-362: “East Germany exhibits larger D(a,t) than observed in West Germany.” Would D(a,t) be affected by sample size (ie, number of locations) besides heterogeneity? Since there are more locations in West Germany than in East Germany, would the direct comparison of D(a,t) be fair?

4. I am not sure about the conclusion that children and seniors had a greater impact than working adults. Essentially what was found was that children and seniors had larger fluctuations in effective reproductive numbers. This does not necessarily lead to the conclusion that children and seniors drived the epidemic stronger than working adults. It might be more appropriate to make this conclusion, if, for example, a study shows that in a mechanistic model (which considers interactions between different age groups), the epidemic dynamics is greatly affected by suppressing the infected number of individuals in children and seniors; or for example, if a study shows that covid cases in children and seniors account for a large fraction of total cases, and the incidence curve is pretty much dominated by children and seniors. 

Author Response

RESPONSE TO REVIEWER 2:

REVIEWER 2:

1. For Equation (3), I believe it is a measure of homogeneity or similarity (ie, the opposite of heterogeneity), because the larger the D(a,t), the less spatial heterogeneity. (When entropy reaches minimum [ie, equal incidence over all locations], D(a,t) reaches maximum, which is 1, right?)

2. Following comment #1, Line 184-185, based on my understanding, should be “equal incidence overall locations gives minimum entropy, hence maximum homogeneity D(a,t)=1”. Or could you please explain if I misunderstood anything?

3. Lines 361-362: “East Germany exhibits larger D(a,t) than observed in West Germany.” Would D(a,t) be affected by sample size (ie, number of locations) besides heterogeneity? Since there are more locations in West Germany than in East Germany, would the direct comparison of D(a,t) be fair?

4. I am not sure about the conclusion that children and seniors had a greater impact than working adults. Essentially what was found was that children and seniors had larger fluctuations in effective reproductive numbers. This does not necessarily lead to the conclusion that children and seniors drived the epidemic stronger than working adults. It might be more appropriate to make this conclusion, if, for example, a study shows that in a mechanistic model (which considers interactions between different age groups), the epidemic dynamics is greatly affected by suppressing the infected number of individuals in children and seniors; or for example, if a study shows that covid cases in children and seniors account for a large fraction of total cases, and the incidence curve is pretty much dominated by children and seniors.

ANSWER:

@1. I agree that the usage of “heterogeneity” and “homogeneity”, respectively, was somewhat confusing. Spatial diversity, D, approaches its maximum 1 with maximum entropy for equal incidences over the the locations, as described in the methods section. Therefore, increasing D means increasing spatial homogeneity. Thus, 1-D obviously describes heterogeneity. In order to avoid cumbersome formulations, it is sufficient to refer to the variation of D to indicate either homogeneity or heterogeneity depending on the context, since it is the relations of D_i and D_j with respect to different regions i and j that matter (particularly East and West). An according explanation in the revised version has been added and the wording has been changed where it appeared appropriate. In particular I changed the corresponding wording in the figure captions and axis labels.

By the way, I recognised that I unintentionally used i as index for both the outer and the inner sum in eq. 2. Thus, I changed to index j for the inner sum.

@2. Maximum entropy gives maximum diversity and is associated with a convergence toward equal incidences among counties, i.e. homogeneity. Please see my comments to response 1. Throughout the revised version the wording has been changed in order to avoid confusion.

@3. Reviewer 2 raises an interesting point in comment 3. Indeed, a dependency of the diversity on the number of locations cannot completely be ruled out, although the denominator, i.e. the number of locations, yields a standardised diversity. I think the problem is similar to an approximation of a continuous probability density by means of a histogram. Shannon entropy is the expectation value of ln(p) and one should expect an effect on the variance due to the choice of spatial granularity rather than on the expectation value itself – at least as long as the granulation is fine enough. Unfortunately, it is still a controversial issue how confidence intervals (CIs) should be constructed for entropies, let alone diversities. Therefore, as described in the text, calculations of CIs have been suppressed in order to avoid mis-interpretations. This issue has now been addressed in more detail in the revised manuscript including an added reference.

@4. Comment 4 by Reviewer 2 addresses indeed to an important issue. However, I feel inclined to defend by position particularly because I raised the suspicion that kids drove the epidemic with all due caution. This issue has been discussed in detail within the Discussion section. Let me paraphrase my position in a perhaps more provocative yet incisive way: I definitely do not conceive kids as drivers in a biological sense. My point is, that the relatively low susceptibility to infections and severe symptoms of kids has been used, if not instrumentalised, as main argument to back political decisions to act irresponsible. Children became the pawn of politicians. The erratic decision-making sequence of openings and closings of schools and other facilities contributed, in my opinion, to the high fluctuation of the reproduction number. In other words, that children are drivers of the pandemic must be interpreted in a passive sense: They were driven to be drivers. Regarding the fact that the share of children in the epidemic was initially underestimated, there is ample evidence, especially in the form of very high seroprevalence rates, cited in the manuscript. The evidence mentioned is of course indirect, but I pointed this out adequately in the discussion. Therefore I ask Reviewer 2 to accept the statement as a hypothesis which is supported by indirect evidence and needs to be examined again in detail in future studies.

Finally, I would like to thank Reviewer 2 for their constructive comments, objections, and recommendations. The Reviewer indeed raised crucial objections which I hopefully addressed in a proper way. I think the quality of the revised manuscript improved considerably. Thanks to the Reviewer for that.

Round 2

Reviewer 2 Report

The author has been responsive to my questions and comments.